# Dynamic Characteristics and Evolution Analysis of Information Dissemination Theme of Social Networks under Emergencies

**DOI:** 10.3390/bs13040282

**Published:** 2023-03-24

**Authors:** Yuan Zhang, Yanxi Xie, Victor Shi, Ke Yin

**Affiliations:** 1School of Management, Shanghai University of Engineering Science, Shanghai 201620, China; 2Lazaridis School of Business & Economics, Wilfrid Laurier University, Waterloo, ON N2L 3C5, Canada

**Keywords:** emergencies, topic dynamic evolution, hidden Dirichlet distribution, online public opinion, life cycle theory

## Abstract

Social media has become an essential channel for the public to create and obtain information during emergencies. As the theme of public concern for emergencies changes over time, there is a lack of research on its dynamic evolution from its latent stage. This paper selects the Henan rainstorm event as a case study and extracts the theme characteristics by combining the life cycle theory and Latent Dirichlet Allocation (LDA) model. It integrates the Term Frequency–Inverse Document Frequency (TF-IDF) and Pointwise Mutual Information (PMI) algorithms as the theme-coding data source to build a dynamic theme propagation model for emergencies. Our research results showed that the theme coding effectively verified the assumption of latent development trends. The dynamic theme model could reveal the theme characteristics of different time series stages of emergencies, analyze the law of the theme evolution of the network’s public opinion, and provide practical and theoretical insights for the emergency management of urban cities.

## 1. Introduction

With the continuous development of network technology, tremendous changes have occurred in how information is released, disseminated, and interacted with. As of October 2022, the latest data show that global Internet users have increased by 3.5% in the past year. The global Internet penetration rate has reached 63.5%. In addition, the number of social media users, which takes up 59.3% of the worldwide population, increased by 190 million in the previous year [1]. It can be seen that social media platforms are flourishing as new channels for information dissemination. Due to its diversified user interaction methods, timeliness, and convenience of information sharing, social media has become an essential space for public expression and information sharing. The two-way or even multiway communication platform built by social media can break the limitation of time, space, and distance in the process of information dissemination [2,3,4] and can achieve the effect of fast and wide communication.

In order to better grasp the golden time after the occurrence of emergencies, gain the initiative, and reduce the severe consequences caused by emergencies, it is imperative to conduct a dynamic evolution analysis on public concern topics on social media platforms under the background of unconventional emergencies, which can provide a theoretical basis for public opinion decision making and analysis. In the literature, researchers divide the whole event into different communication cycles and summarize the public opinion themes in different periods [5,6,7] and finally give relevant policy suggestions or predict the event through modeling. In this way, dividing events into independent research often weakens the dynamic evolution process when the event development process is understood, and the event is studied from the time node after the emergency, meaning the focus is on how to avoid risks when similar events occur in the future, and exploring unconventional emergency events is not concentrated on. Solomon et al. [8] note that it is difficult to stop a pandemic when it breaks out, and the best way to do so is to identify danger signs as soon as they appear. And, the dynamics of social information in the early stages provide a critical perspective. Therefore, tapping into the potential of emergency events, preparing accordingly in advance, and reducing the damage caused by emergency events is a new perspective worth studying.

This paper starts with the latent development time node of unconventional emergencies, takes the recent “Henan rainstorm” urban natural disaster event as an example, analyzes microblog texts in combination with the life cycle theory and hidden Dirichlet model, and explores the topic of dynamic evolution information in the background of the sequence, whereby we mine the evolution law of public attention and put forward suggestions for improvement. Conducting a public opinion evolution analysis can show the discussion hotspots of the network’s public opinion at different stages in the propagation cycle of unconventional emergencies and the dynamic trend changes over time. By counting the number of popular microblogs in the “Henan rainstorm” incident hour by hour, it was found that it took a shorter time to reach the breaking point of public opinion than before, so the advent of the new media era requires that we must plan in the face of public opinion crises. It can help guide public opinion on the Internet when dealing with similar incidents in the future, which will provide a theoretical basis and decision support.

### 1.1. Social Media and Emergency Management

With the development of Internet technology, social media has become a popular form of media, and it provides new channels for information sources as well as rapid communication [9]. Social media platforms contain diverse types of information and provides researchers in various fields with diverse research ideas, such as using geotagging data from social media platforms to evaluate the performance and robustness of methods to detect individuals’ basic locations [10]; using location-based social network data to provide innovative ideas to classify cultural tourist attractions and understand the preferences and needs of tourist groups [11]; and using social media data to analyze the perceptual characteristics of investor sentiments and stock correlations and construct nonlinear models to complete stock price predictions, which have been conducted by some researchers [12] in the financial field.

Because social media facilitates the public to publish information in different forms (text, images, videos), it has an important place in the field of crisis management [13]. Social media is a valuable source of data in the context of a crisis [14] as it provides the conditions for the needs of different groups to be met. For example, during emergencies, social media communication is a way for the public to rely on and participate in seeking and providing information and social support [15]; emergency responders use social media to obtain information and study public reactions to reduce damage and provide possibilities to improve relief procedures [16,17]; relevant management can use social media to improve awareness of emergencies in real time; and the public in the incident area can provide almost-real-time disaster-scene information through the network, which can provide a strong basis for decision making in terms of how to make a rescue action plan [9]. Social media information has big data characteristics in the four V’s: volume, veracity, variety, and value [14], which also makes the analysis of social media information highly complex. Stieglitz et al. [18] used empirical cases to analyze comments, or text-based social media communication, to better understand the intersubjective perception processes and patterns in extreme events. Li et al. [5] constructed the SIR epidemic model to simulate the interaction among three groups of Internet users, media, and government and selected mainstream social media data as the entry point to classify the emergencies into four categories for empirical study; the quantifiable results generated from their research were very effective at managing the public sentiment of public opinion. Based on the example of Hurricane Irma, Zhang et al. [7] explored the factors that influenced the dissemination of information in the social media environment, and the findings provided a practical basis for crisis response management. Li et al. [19] explored the influence of emotional factors on a communication scale based on information related to natural disaster events on social media platforms, which could help emergency managers develop emotional strategies after disaster events. Pohl et al. [13] used clustering algorithms to identify crisis-related subevents based on social media data, which is important for optimal crisis management.

In summary, we can understand the important role played by social media in dealing with emergencies, and its rich data provide ideas for diverse research. In this paper, we select data from social media platforms to study the evolution of public attention themes in emergencies and provide theoretical support for emergency management.

### 1.2. Life Cycle Theory

The “life cycle theory” comes from the evolutionary process of organisms in biology, which change the organisms’ appearance, shape, and function accordingly and show stages. Levitan [20] argues that information or information resources, as a special commodity, also have life-cycle characteristics. Online public opinion is the sum of diverse public expressions (opinions, emotions, discussions) about public events in a specific period of time, and the essence of the communication process is to use information as a carrier, so it also has a complete life cycle as well as different stage characteristics. Therefore, based on the life cycle theory, researchers often divide the whole process of an opinion event from its creation and further disseminate it until its demise into different stages. The evolution of online public opinions is combined with the life cycle theory, which divides an opinion event at its generation and disseminates it into different stages until extinction. The evolution process of unconventional emergencies often presents a regular pattern. Due to the different starting points and measurement standards of previous researchers, the division results based on the life cycle of different unconventional emergencies are also different. There are three stages [5], four stages [6], and five stages [21]. Burkholder and Toole [22] propose a classic three-stage propagation model. Liu et al. [23] take the “Daliangshan” fire in Sichuan, China, as the research subject and integrate the life cycle theory to propose a public opinion communication model in three stages (formation period, outbreak period, and recession period) and explore the law of the network’s public opinion communication under the background of disasters by analyzing the concerns of different communication periods. Chen et al. [24] conducted empirical research based on social hotspot events and divided the development of public opinion into six stages according to the "bimodal characteristics" of public opinion evolution. Li and Shen [25] constructed a social network public opinion communication model and applied it to the field of animal epidemics and combined the life cycle theory to divide the communication cycle into three stages. Zhang et al. [6] divided the Zika virus event into four time periods, selected relevant topic discussions on popular social media platforms as research data, and explored the evolution of the theme. This article divided the development stages of public opinion by sorting out relevant research and combining the development characteristics of the event itself with the life cycle theory.

### 1.3. LDA Topic Discovery

The LDA (Latent Dirichlet Allocation) model is a classic Bayesian model and includes a three-layer structure of documents, topics, and words. It is a probabilistic generation model used to mine potential topics in texts, as shown in Figure 1. The occurrence probability of each word in the document generated by the LDA model is shown below:(1)$$P\left(\mathrm{word}/\mathrm{text}\right)=\sum_{\mathrm{theme}}P\left(\mathrm{word}/\mathrm{theme}\right)\times P\left(\mathrm{theme}/\mathrm{text}\right)$$

The topic model of the LDA model is developed based on the PLSA model. The calculation of the original model is complicated by introducing the Dirichlet prior distribution. LDA is one of the most commonly used text topic mining models. Compared with traditional text semantic analysis, LDA has a better effect in analyzing large-scale unstructured data sets [26]. The data collected in this paper were from the Sina Weibo platform. The selection of popular microblogs can effectively avoid the shortcomings of the LDA model in short text analysis and make the results more accurate.

In the study of opinion topic models, LDA topic models combined with temporal dimensions used to mine the evolution pattern of topic content are commonly used by researchers. Combining other complex models with LDA models to mine text information and change patterns in different dimensions is also a popular research direction. Chen et al. [24] used the LDA model to find and cluster topics at each stage, calculate topic similarity using cosine similarity, and construct a multiagent public opinion analysis model based on the perspective of multidimensional stakeholders. Feng et al. [27] employed the LDA model to extract characteristic words in different time series stages and combine word similarity and cooccurrence relationships to build a topic attention evolution model and verify the effectiveness of the model with the help of real data sets. Based on the improved BERT model and LDA theme model, Tan et al. [28] added theme dimensions to the emotional analysis and verified the effectiveness of the mixed model at predicting public opinion and emotional evolution through an empirical analysis of hot social events. Based on the LDA model, Xie et al. [29] mined the theme expression of the public’s opinion on the COVID-19 pandemic that was expressed on social media platforms.

To sum up, the LDA model is one of the most effective methods used to mine text topics. It has gained significant interest from researchers who are interested in modifying LDA models in order to improve the accuracy of topic mining. However, previous researchers adopted the method of dividing events into different periods and analyzing the characteristics of each stage, which ignores the dynamic evolution process in the development cycle of public opinion events and weakens the connection between different periods. Since there is little research on the incubation period before an emergent event, this paper chooses a recent emergent event with great impact as an example, applies the life cycle theory and the implicit Dirichlet distribution theme model, excavates the theme characteristics of the different stages of public opinion dissemination, and constructs a dynamic theme evolution model through user-defined theme codes to explore the theme dynamic evolution law in the overall process of the event. In order to help deal with similar events, the effective guidance and control of online public opinion provide a theoretical basis and decision support.

## 2. Materials and Methods

### 2.1. Research Design

This study takes the recent unconventional emergency event of “Henan rainstorms” as an example and selects “Sina Microblog” as the data-source platform. In China, microblogs are an important Internet space for public information expression [23] and are an essential database for researchers.

The Henan rainstorm incident was a high-impact event that attracted widespread attention and could thus provide a rich data source for this study. Moreover, regarding the thematic evolution of the incident, it can be found that before the heavy rainfall in Zhengzhou caused significant damage, heavy rainfall extremes had already occurred in neighboring cities with severe consequences. However, because the city is inland, the emergency services personnel lacked experience in managing heavy rainfall disasters, which eventually led to severe consequences. Therefore, the study of heavy rainfall events in Henan Province can provide a reference for other inland cities and a theoretical basis for crisis management intervention points.

The initial corpus was built by searching keywords and analyzing the texts of “popular tweets” related to the events. The overall research framework is shown in Figure 2, which is based on the life cycle theory and the LDA model to dynamically mine topics at different stages of the event communication cycle. First, we searched for the keyword “Henan rainstorm” to explore the relevant popular microblog texts and preprocess the collected texts, including Chinese word separation, deactivation, and adding user-defined dictionaries. Then, we divided the public opinion dissemination cycle into stages based on the life cycle theory and event development curve. We then carried out LDA topic modeling for each stage of the microblog corpus to extract relevant topic words. Finally, user-defined topic coding was applied to the microblog corpus to explore the dynamic evolution of related topics in the whole life cycle. The results of the LDA topic word extraction were combined to perform hot topic mining and a dynamic topic evolution analysis.

### 2.2. Public Opinion Stage Division

Based on the division of the evolution stages of online public opinion in the existing literature and the characteristics of the real data set, this paper divides the process of public opinion dissemination into four stages: the incubation period, explosion period, recession period, and stabilization period based on the life cycle theory.

First of all, the incubation period refers to the stage when public opinion is just occurring along with the emergence of events, the number of topics related to public opinion is small, the attention is partially great, and the event fails to attract widespread attention.

In addition, the explosion period refers to the period when the amount of information related to public opinion grows rapidly and reaches its highest point, the attention to events increases comprehensively, the number of public opinion topics becomes diversified, and the discussion reaches its peak.

Furthermore, the recession period refers to the situation where the heat of the matter decreases rapidly after the explosion period. Still, a small-scale rebound of the derived public opinion may occur.

Finally, the stabilization period refers to the basic end of the development of the matter and the stabilization of the online public opinion, with the amount of discussion and attention related to the matter showing a decreasing trend.

### 2.3. LDA Model Parameter Setting

Since the number of topics in the LDA model needs to be set artificially, this paper selected the confusion index to judge whether the selection effect of the number of topics was optimal. Based on the comprehensive performance of the perplexity curve in four periods, the final number of topics was selected as 8. In addition, the selected number of topics achieved a good differentiation effect and did not overfit. The confusion curves for the four periods are shown in Figure 3a–d. The formula for calculating the confusion degree is shown in Equation (2). The selection of the number of topic words needs to be discussed based on specific events. Chen et al. [24] selected the top 20 theme words with the highest probability of occurrence to reflect the thematic characteristics of hot social events in different periods. Feng et al. [27] chose the top 10 words as the characteristic words of the subthemes in each time slice. Xie et al. [29] selected the top 30 most salient words to be displayed visually as theme words. In this paper, when determining the number of topic words, 30 topic words were finally selected. The number of iterations was set to 300, and the hyperparameters were set to α = 0.01 and β = 0.05. After the parameters were set, the microblogging text corpus was input, and the LDA topic model program was run.
(2)$$\mathrm{Perplexity}\left(D\right)=\exp\left\{-\frac{\sum_{i=1}^{M}\log p\left(d_{i}\right)}{\sum_{i=1}^{M}N_{i}}\right\}$$

### 2.4. TF-IDF Algorithm and PMI Algorithm

The TF-IDF algorithm is commonly used to extract keywords due to its feature of fusing word frequency and word importance. The calculation formula is as follows:(3)$$\mathrm{TF}=\frac{\mathrm{Number}\;\mathrm{of}\;\mathrm{occurrences}\;\mathrm{of}\;a\;\mathrm{word}\;\mathrm{in}\;\mathrm{the}\;\mathrm{article}}{\mathrm{Total}\;\mathrm{number}\;\mathrm{of}\;\mathrm{articles}}\;$$
(4)$$\mathrm{IDF}=\log\left(\frac{\mathrm{Total}\;\mathrm{number}\;\mathrm{of}\;\mathrm{documents}\;\mathrm{in}\;\mathrm{the}\;\mathrm{corpus}}{\mathrm{Number}\;\mathrm{of}\;\mathrm{documents}\;\mathrm{containing}\;a\;\mathrm{word}+1}\right)\;$$
Value of TF−IDF=TF∗IDF

PMI (Pointwise Mutual Information) can be used to calculate the similarity between two words. The formula is as follows:(5)$$\mathrm{PMI}\left({\mathrm{word}}_{1},{\mathrm{word}}_{2}\right)=\log_{2}\left[\frac{p\left({\mathrm{word}}_{1}\&{\mathrm{word}}_{2}\right)}{p\left({\mathrm{word}}_{1}\right)p\left({\mathrm{word}}_{2}\right)}\right]$$

## 3. Results

### 3.1. Research Data Collection and Processing

One of the most used and influential user interaction platforms in China is the Sina Weibo platform. The microblogging platform was an important channel for public expression and information exchange during the Henan rainstorm outbreak, and it provided a sufficient data volume for this research. In this article, Sina Weibo platform data are used as the data source, and the keyword “Henan rainstorm” was used to explore the popular microblogs from 25 June 2021 to 30 August 2021, and a total of 11,747 data were collected. The data of the Henan rainstorm event were selected from June 25, which is in line with the purpose of mining the latent trend before the event.

In order to improve the accuracy of the experimental results, text preprocessing work was performed on all the popular microblogs used, including the steps of jieba word separation, removing deactivated words, and adding user-defined dictionaries. In the process of the experiment, the user-defined dictionary and the deactivated word list were continuously updated and optimized according to the judgment of the model output until the output was relatively satisfactory.

### 3.2. Stage Division of Event-Related Public Opinion Dissemination

The time series distribution of popular microblogs related to the “Henan rainstorm” is shown in Figure 4. Based on the trend of "single peak" in the number of popular microblogs, this paper divided the cycle of public opinion dissemination into four phases: the incubation period (25 June 2021 to 19 July 2021), the explosion period (20 July 2021 to 24 July 2021), the recession period (25 July 2021 to 2 August 2021), and the stabilization period (3 August 2021 to 30 August 2021).

### 3.3. Topic Dynamics Mining in the Communication Cycle of Unconventional Emergencies

The LDA themes were extracted from the corpus of the microblogging texts in the incubation, explosion, recession, and stabilization phases. The final results of the eight topics in each phase and the related feature words of each topic were obtained. In addition, five topic words with high relevance under each topic were selected for theme interpretation. The criteria for selecting highly relevant words were a combination of word probability and topic discrimination in the LDA model results. However, after several experiments, it was found that when only the word probability value was considered when selecting some of the topic features, the top-ranked words in some topics may have been duplicated, which was not conducive to showing the topic differentiation. Therefore, the five words with high relevance that were selected by considering the word probability and topic variability are shown in Table 1, Table 2, Table 3 and Table 4.

#### 3.3.1. The Incubation Period

Table 1 shows that many overlapping discourse words appeared in the first four topics of the incubation period, which could be combined into the same topic—"weather warning.” The occurrence of location words (such as “North China”) indicates that many geographic location words appear in the early warning messages of rainstorms to alert the public in the corresponding areas of the occurrence of rainstorm disasters. Words such as “rainstorm” and “strong convection” frequently appear in the first four topics, conveying the message to the public that extreme weather is coming and needs to be paid attention to and prepared for in advance. In addition, the words “continue posting” and “China Central Weather Station” indicate that the weather forecast information comes from official channels, and the warning information issued is continuously updated according to the latest weather changes, reminding the public to take precautions. The words “go out”, “must” and “caution” indicate that the official reporting of weather conditions will be accompanied by corresponding safety reminders to the public, who can prepare for rainstorms in advance to reduce the damage that may occur. On the other hand, the appearance of a large number of prefecture-level city names (such as “Liyuan” and “Jiaozuo”) can be observed. By sorting out the events, we can draw the following conclusions: on the one hand, the weather forecasts provide early warnings of possible rainfall situations in the prefecture-level cities; on the other hand, some of the surrounding prefecture-level cities had already experienced rainstorm disasters before the occurrence of the extraordinarily heavy rainfall in Zhengzhou, which echoes the appearance of theme words such as “rescue”. From the combination of words such as “scenic spot” and “flood control” in Theme 4 and “reservoir” in Theme 3, it is clear that in the case of continuous heavy rainfall, scenic spots with major safety hazards should be closed in time and tourists in the spots should be evacuated safely. Residents in mountainous areas should pay attention to prevent disasters such as flash floods. The public is very concerned about topics such as continuous rainfall causing a large number of reservoirs to exceed the flood limit level.

Themes 5–8 in Table 1 all repeatedly included words related to “rescue” and “firefighter,” as well as specific names and personal pronouns (such as “man” and “man in white”). Considering the similarity of the topics, the above four topics can be grouped into the theme of “warm-hearted moments”. In the context of the rainstorm disaster, whether it is the professional rescue by firefighters or the rescue by brave ordinary people, these topics spread rapidly through the news media’s rescue reports and become the focus of public attention.

As mentioned above, the division of the text data topics in the latent period is relatively clear and simple as it is mainly divided into two major parts: “weather warning” and “warm-hearted moments”, and it can be seen that the words related to “rainstorm” appear many times in each division. This can be explained by the fact that the weather changes quickly, and thus the weather forecast information is swiftly updated, which shows that the weather forecast department can effectively warn of possible disasters before the arrival of heavy rain.

#### 3.3.2. The Explosion Period

As shown in Table 2, the words “danger avoidance” and “self-help” appeared in topic 1, indicating a new change in public concern. The topic of how to protect oneself when a rainstorm comes is on the rise, and there are more and more science-knowledge posts on online media. The words “flood”, “food”, and “infectious disease” show that many netizens are keen to remind others about possible secondary disasters.

As shown in Table 2, keywords such as “mutual help” and “help-seeking” contained in topic 2, topic 4, and topic 5 showed that after the storm disaster occurred, the public released a lot of help information with the power of online media; the information appealed to netizens who could forward and spread the information to increase attention to stranded people in need of help, and Internet users also showed images of the more seriously affected areas to the public through the Internet.

Topic 6 in Table 2 mainly revolved around the words “donation” and “charity”, showing that after the storm disaster, forces from all over the country converged on Henan, enthusiastic netizens donated to Henan through public welfare platforms, and enterprises and individuals rushed to help Henan by performing any actions they could.

Topic 7 shown in Table 2 revolved around the theme word “rescue”, and the theme expressed was still around the topic of “warm-hearted moments”. The names of provinces such as “Anhui” and “Jiangsu” appeared in the topic word, indicating that after the heavy rainfall disaster in Henan, the national firefighting force rushed to Henan and went to different affected prefecture-level cities to carry out rescue work.

As shown in Table 2, the appearance of words such as “typhoon” in Topic 8 revealed the cause of the rainstorm, and combined with words such as “extreme” and “rare”, they are enough to reflect the unusual nature of this rainstorm. There are still a lot of words related to the theme of “weather warning” for this topic, so it can be summarized as this theme.

To sum up, the text topics in the explosion period were more diversified, with new themes of “help-seeking information” and “relief donation” appearing as the event developed in addition to the themes of “weather warning” and “warm-hearted moment” that already appeared in the incubation period.

#### 3.3.3. The Recession Period

As shown in Table 3, the presence of words such as “missing” and “die in an accident” in Topic 1 showed that the development of the event to that stage resulted in an official summary website that informed the public about the casualties caused by this rainstorm. Topic 2 and Topic 6 existed with different occupational names, specific personal pronouns, etc., all of which showed that in the course of the rainstorm disaster, many ordinary people held on to their jobs. Many ordinary people stepped forward and bravely saved people, and these warm stories were widely disseminated through news media reports and attracted wide attention. The names of different provinces appeared in Topic 2, which echoed some of the topic words in Topic 7 during the explosion period considering that there was a certain lag in the dissemination process of the news reports. The corresponding number of topic words was compared with the explosion period, which was in line with the law of event dissemination.

The words “assault boat” and “rainstorm mutual aid “ in Topic 3 in Table 3 showed that after the storm disaster, various emergencies occurred such as city flooding, hospital power outages, the reservoir water level exceeding the limit, and water appearing in subway entrances; in response, the public spontaneously formed a help network through the Internet, updated help information, and supplied needed materials in real time so that more people could carry out effective help.

As shown in Table 3, there was a causal relationship between many of the topic words in Topic 4 and Topic 5, which could be summarized as a typhoon-induced rainstorm, and many aspects of urban life were affected by the rainstorm as a result.

In Topic 7 and Topic 8, as shown in Table 3, there was a lot of overlap in terms of thematic terms, mainly around the theme of “relief donations”. It can be seen that the topic words were richer than during the explosion period. As public figures, celebrities donated money after the rainstorm and drove their fans to donate through their own influence. There were also many national brands that were taking initiative to assume corporate social responsibility and contribute to Henan.

To sum up, although there was no noticeable increase in the theme of the recession period, the content of the theme words was richer. Whether it was “relief donation” or “help-seeking information”, they all showed more diversified information, which was related to the wide dissemination of news reports with a certain lag, which is in line with the law of information dissemination.

#### 3.3.4. The Stabilization Period

Many words in Topic 1 in Table 4 revolve around “investigation”, which shows that the development of the incident gradually subsided and entered a new stage in which the investigation of the heavy rainfall in Zhengzhou was conducted to give an explanation to the public. Topics 2 to 7 can be summarized as the same theme that mainly revolves around the two themes of “weather warning” and “safety tips”. After the heavy rainfall in Zhengzhou, the public was alerted to the new round of heavy rainfall and prepared in advance, such as by preparing sandbags or buying food. Supermarkets were prepared in advance and learned from the experience of previous rainstorms, which resulted in a large number of cars being soaked. Hence, a large number of private vehicles were parked early in the higher ground area, and for such exceptional circumstances, the traffic police department also responded by not giving out punishments.

To sum up, the topic of the text in the stabilization period was relatively simple, the development of the event entered a calm state, and the heat surrounding the new round of rainstorm warnings decreased; additionally, by considering all of society’s responses as well as the gradual subsidence of the rainstorm event, the investigation of this rainstorm disaster event was conducted in an orderly manner to give the public an explanation of the event.

### 3.4. Dynamic Evolution of Theme in the Public Opinion Communication Cycle of Unconventional Emergencies

The LDA model tended to obtain information about the topics contained in the text corpus at a certain stage and the characteristic words within different topics. Some themes may recur in the whole life cycle (for example, topic 1 in the incubation period and topic 8 in the explosion period both mainly express the weather warning aspect). But even similar themes had different attention levels in different periods, and their changes could reflect the shift in public attention, which is important when mining the dynamic evolution characteristics of themes.

In this paper, the topic hotness value was calculated using topic coding, which is a set of coding rules containing several related feature words. For example, the weather warnings topic contains topic feature words (early warning, Chinese Central Meteorological station, etc.) and the surrounding cities topic contains topic feature words (Nanyang, Xinyang, Xinxiang, etc.). The encoding rules can be used to capture keywords for the whole life cycle of the text corpus and thus obtain the hotness values of related topics in different periods. This paper integrated TF-IDF keyword extraction and a PMI (Point Mutual Information) algorithm to draw a word co-occurrence network as one of the data sources for constructing topic codes, while the results of the topic words extracted by the LDA model were also used as a supplementary word bank so that the integrity of the topic feature word selection could be effectively realized and hot topic mining and a dynamic topic evolution analysis could be carried out. In order to make a strong distinction between different topic codes, when selecting topic words of different topic codes, one should avoid the selection of possible interference words. For example, the interference word “rainstorm” should not be selected for the topic of help-seeking information.

This paper drew a network diagram (Figure 5) that was initially divided into five modules according to the community detection algorithm, but when taking into account factors such as the results of more refined studies on related topics and the latent period surrounding city warnings, the theme word module was finally determined to contain six related topics (weather warnings, surrounding cities, safety tips, warm-hearted moments, help-seeking information, and relief donations) after combining the word network theme results for several optimizations of the theme word module.

The results of the heat map (Figure 6) and bubble map (Figure 7) show the overwhelming amount of “weather warning” information in the incubation period. The proportion of “surrounding cities” and “safety tips” is similar, which echoes the extraction of the topic words and indicates that a large number of rainstorm warnings have already appeared in the weather forecasts of neighboring cities. Combined with the fact that the proportion of “warm-hearted moments” was more than 1/3, it can be seen that before the occurrence of Zhengzhou’s heavy rainstorm, the surrounding cities had already experienced heavy rainstorm disasters, and fire rescue and other information dissemination through news media reports occupied an important proportion of the incubation period. To sum up, before the occurrence of the heavy rainstorm in Zhengzhou, disaster events caused by heavy rainstorms had already occurred in the neighboring cities. By analyzing the information from the incubation period, we can look for potential possibilities before the occurrence of an emergency and respond effectively to reduce the possible losses caused by the emergency.

Compared with the incubation period, the topic evolution of the explosion period showed a “diagonal” state, with the proportion of “help-seeking information” and “relief donation” themes increasing rapidly, which was in line with the theme feature word extraction. This indicated that the public released a large amount of unofficial mutual help information through the Internet after the rainstorm and that the rainstorm in Henan attracted widespread attention within a short period of time. The rapid increase in the proportion of the “relief donation” theme showed that the unity in the blood of the nation is stimulated in the face of sudden disasters, and the whole nation was united to help Henan.

Compared with the explosion period, the proportion of “help-seeking information” in the text topics in the recession period decreased, and the keywords in the user-defined dictionary were mainly divided into two aspects: real-time help information and postdisaster urgent supplies. In the explosion period, the proportion of real-time help information was larger, while in the recession period, the theme of “help-seeking information” mainly focused on the information of the materials needed after the disaster. The proportion of “warm-hearted moments” kept increasing and reached the top value in the recession period, which was consistent with the extracted results of the theme words due to the repeated nature of the rainstorm and the delay of news reporting.

After entering the stable period, it can be seen that except for “Safety tips”, all the other topics were rapidly declining in popularity, and the development of the event entered a stable stage with intermittent small climaxes due to the repetitive nature of the heavy rainfall. The combination of “weather warning” and “safety tips” showed that the public became more alert to the new round of rainfall after the heavy rainfall in Zhengzhou, and all relevant departments were fully prepared to face the possible situation. Relevant departments should pay timely attention to the impact of the rainstorm disaster on all walks of life. For example, as the event entered a stable period at the end of August, schools faced the opening of schoolwork, postdisaster reconstruction, the resumption of work and production, and other issues, which required the introduction of relevant measures.

In summary, the user-defined topic codes could clearly show the changes in the proportion of different topics in the whole life cycle of event development, tap the patterns between different periods, and better show the hot trends of discussions at each stage of the public opinion dissemination cycle in the context of sudden natural disaster events, which can help relevant management departments give early warning tips when dealing with similar events in the future and can provide a theoretical basis and decision support to guide online public opinions.

## 4. Discussion

In the era of rapid Internet development, information breaks the traditional channels of information dissemination with the help of social media platforms, which revolutionize different aspects of its generation, dissemination, and reception [30]. Social media platforms have a mediating effect on information dissemination, which not only enables the rapid spread of emergencies but also provides the possibility of active public participation [23]. The development of diversified public expressions has contributed to the advent of the era of self-media, which provides convenient conditions for every citizen to express their opinions and emotions. The way in which the social media space empowers the individual user is unprecedentedly revolutionary compared to the traditional environment restricted by mainstream media [31]. However, users are easily influenced by subjective emotions and one-sided information, which leads to misunderstandings and dissemination and increases the risk of online rumors. The number of popular microblogs in the “Henan rainstorm” incident has been counted hour by hour, and it was found that the outbreak point of public opinion is reached in a shorter time than before.

### 4.1. Government

For the government, the timely handling of relevant public opinion is the best solution. After the occurrence of an emergency event, the government should be the first to promptly release the progress of the event through an official platform, constantly update the information, carry out the investigation and analysis of the event, update the causes of the emergency event and the progress of the investigation, and listen to the public opinion to ensure that the dissemination of information is not one way [23]. The publishing mode with fast frequency and high density reduces the probability of a public opinion crisis caused by information asymmetry [32]. Effective information aggregation that utilizes established special accounts or websites is used to avoid the severe consequences of misleading the public through online rumors or fermenting public opinion with emotions. At the same time, timely updates on decision making and emergency measures through cooperation with the media can keep the public accurately informed of the progress and defuse public sentiments [5]. The results of extracting the theme of “heavy rainfall in Henan” in the budding period showed that heavy rainfall had already occurred in other neighboring cities in Henan before the occurrence of the heavy rainfall in Zhengzhou. Still, because Henan is located in a plain area and has little experience in facing heavy rainfall, it did not attract attention in the budding period. After a sudden event, the government should take emergency measures to guarantee the continuity of essential services, such as electricity, water, and other utilities that are very important during a disaster [33,34]. Moreover, the government should take timely measures to coordinate the work of relevant departments to avoid tragic events. When the flood season is approaching, relevant departments should make timely preparations and provide comprehensive warnings through the Internet, cell phones, and public electronic screens to remind the public of the need to be alert to extreme weather in the near future. In addition, we can sense that the government should play a more critical role in urban planning to make cities and communities environmentally friendly and provide safer, healthier, and more comfortable living conditions for all citizens [35].

### 4.2. Enterprises

Nowadays, posting on social media platforms is the way that companies promote their brands, products, and corporate images [36]; thus, these platforms have become a tool for effective interactions between companies and users [37], and they enhance information sharing between companies and users [38] while reducing communication costs. The high popularity of social media has made it widely recognized as a critical strategic component for organizational competitiveness and survival [39,40]. For enterprises, taking on corporate social responsibility is a crucial way to gain word of mouth. Regarding the “Henan Rainstorm”, there were many such enterprises that took on social responsibility, including Henan’s local brand of “Mixue Ice Cream & Tea” (a Henan Local Famous Milk Tea Brand) that made warm-hearted donations, as well as “ERKE” (a Chinese sports brand) and “Huiyuan Juice” (a Chinese juice brand) that made significant contributions despite the great difficulties in their operations. These companies have interpreted what it means to resist disasters with one heart through their actions. Their active CSR behaviors have been widely spread on social media platforms, which has resulted in a large amount of positive electronic comments [37]. A good word of mouth is one of the most important drivers of consumer attitudes and behaviors [41]. It not only triggered a wave of national product boom in the whole society but companies also took the opportunity to increase their visibility and enhance their image in the public mind, which gave them a chance to grow again.

### 4.3. News Media

The media plays a vital role as a link between the government and the public, and the timeliness of news is its value. In the face of emergencies, the network will be flooded with a large number of reports on emergencies. The news media should make a rigorous and authentic report on the event itself and continue to follow up on the event’s progress. The news report itself enables the public to more easily understand the event through the Internet, so the news report should uphold a true and rigorous attitude. All information should be based on the actual situation of the event. Therefore, news reports should keep a thorough attitude, and all information should be based on the actual development of the event; eye-catching news reports that simply draw attention should not be written. Fake news is a burden on contemporary society and has a significant impact on citizens, countries, and organizations [36].

In particular, fragmented reading has become a popular way to occupy our lives in the age of self-media. Short videos and articles can attract a lot of attention through eye-catching titles, but such a format may be accompanied by the covert spread of online rumors. The public interprets things through fragmented texts and videos and adds subjective understandings and emotions for secondary dissemination, which gives favorable conditions for the fermentation of online rumors. Therefore, news media should be rigorous in reporting based on facts.

From the empirical analysis of theme mining, it can be seen that the theme of the needs of the public after an emergency is divided into different modules. During the rainstorm in Henan, rainstorm mutual assistance became one of the crucial unofficial help platforms for the public. Through the explosive release and reprint of information on social media platforms, there will inevitably be a duplication of information and failure to update the rescued information promptly, which will result in a waste of rescue efforts. In media platforms, this process should be improved from the perspective of user needs. Social media platforms can establish an official topic channel of the media to summarize information and update it promptly to avoid the resource waste caused by information redundancy.

By analyzing the dynamic evolution process of the event theme, we could mine the change rule of public opinion development and the change in hot topics in different periods from the complex data, which could thus help the relevant management departments improve their efficiency and decision support when dealing with similar events in the future.

## 5. Conclusions

The rainstorm event in Henan that was selected for this study was of practical research significance. The rainstorm event in Henan reflected the lack of experience of inland cities in natural disaster emergency management. Ecological and environmental issues are the key concerns of today’s society, and we should learn from the experience of the Henan rainstorm event to provide ideas for the emergency management of other inland cities. The dynamic evolution process of theme mining and the theme hotness value can provide theoretical support for relevant managers to understand public needs and grasp the changes in public needs to improve emergency relief work. The multiperspective discussions among enterprises, media, and the government can enable different participants to gain experience and improve their work. The theme of “surrounding cities” was used to verify the assumption of the existence of latent phase crises and to enrich emergency management research. By combining the life-cycle theory and the LDA theme modeling, this study mined and analyzed the themes of texts at different stages of the communication cycle and explored the dynamic evolution of themes in the development of public opinion by setting user-defined theme codes and mining the hot themes at each stage. In terms of theory, this study combined the LDA theme mining model and life cycle theory to realize the integration of theory into practice. Previous studies have mainly analyzed each stage of public opinion dissemination independently. Still, this paper established user-defined topic codes based on the whole corpus collected. It reflected the dynamic evolution of various topics in the life cycle of public opinion dissemination more accurately and intuitively through visualization results, which provided relevant ideas for public opinion topic mining and evolution analysis.

In future research, we will consider collecting data from various platforms, such as the popular short-video platform TikTok, where videos give users a richer sensory experience. Still, the update of communication media will also lead to an increase in the risk of emotional public opinions on emergencies. Meanwhile, we need to enrich the selection of breaking news cases to improve the practical significance of our findings. We will break the research boundaries of different influencing factors by constructing an opinion network and linking user nodes with themes and emotional factors to study the impact of multidimensional online public opinions.

## Figures and Tables

**Figure 1 behavsci-13-00282-f001:**
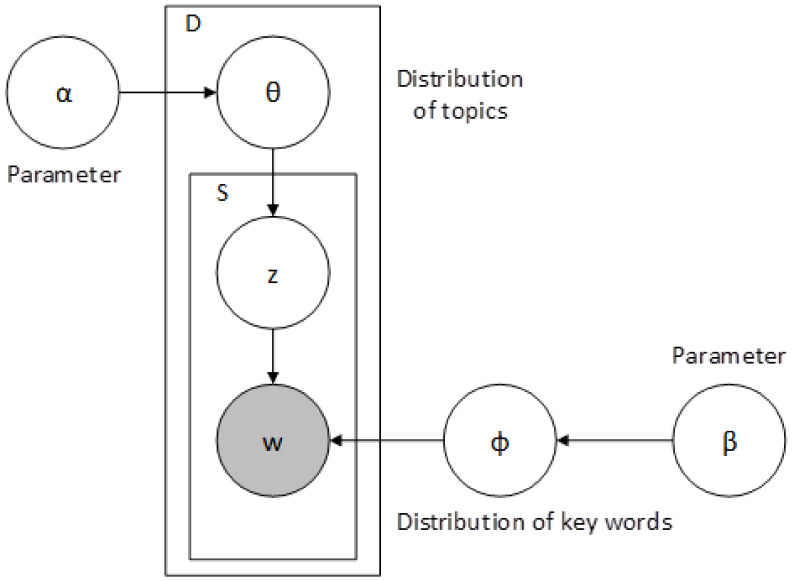
The LDA model.

**Figure 2 behavsci-13-00282-f002:**
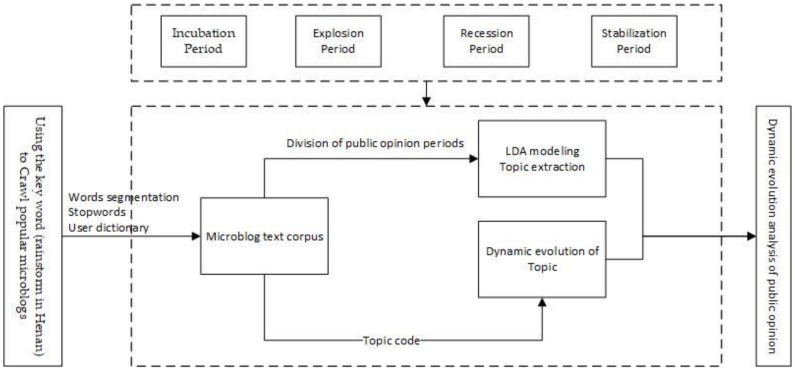
The research Framework.

**Figure 3 behavsci-13-00282-f003:**
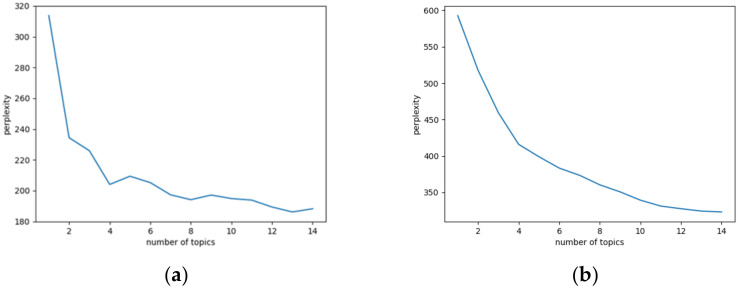
Perplexity graphs. (**a**) Incubation period; (**b**) explosion period; (**c**) recession period; (**d**) stabilization period.

**Figure 4 behavsci-13-00282-f004:**
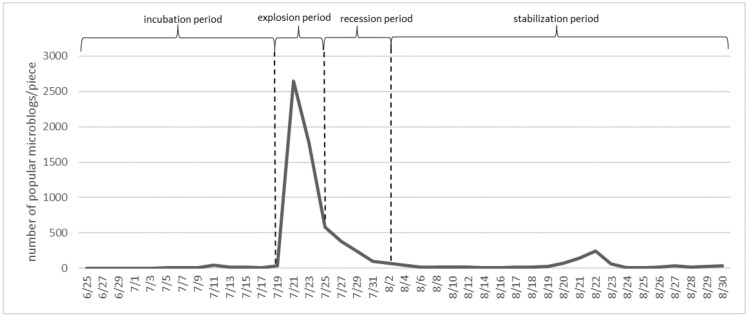
Public opinion event popular microblog trend.

**Figure 5 behavsci-13-00282-f005:**
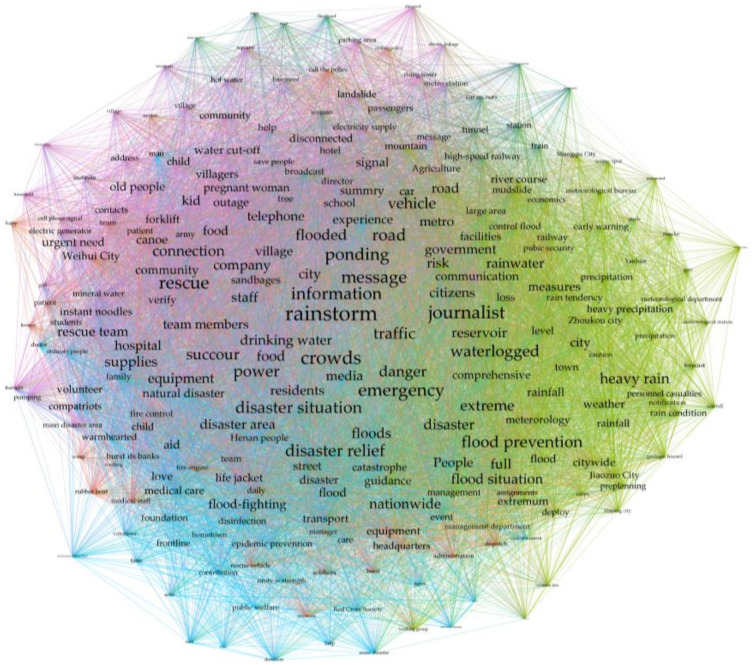
Initial module division diagram of the first 200 keywords in co-occurrence network.

**Figure 6 behavsci-13-00282-f006:**
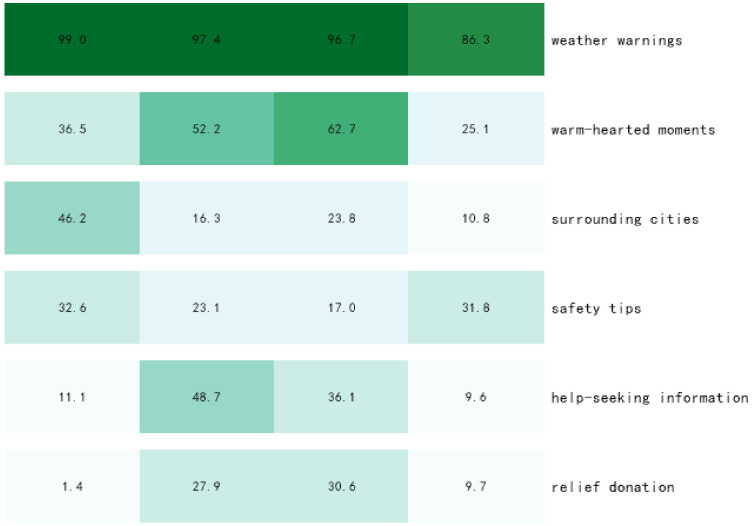
Distribution of thematic heat map of public opinion information dissemination cycle.

**Figure 7 behavsci-13-00282-f007:**
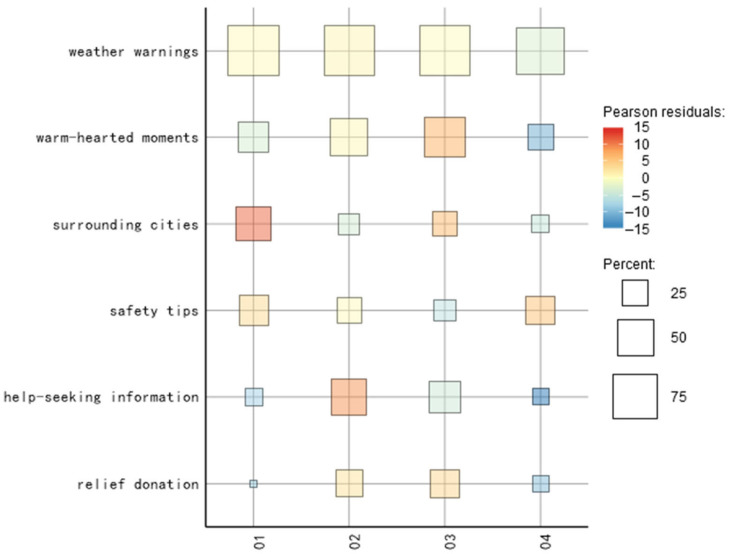
Bubble chart distribution of public opinion information transmission cycle.

**Table 1 behavsci-13-00282-t001:** Text topic extraction results in the incubation period.

Incubation Period					
Topic 1	rainstorm	Early warning	Strong convection	Continue posting	Chinese Central Meteorological Station
Topic 2	remind	go out	must	Caution	Rainstorm blue warning
Topic 3	North China	reservoir	risk	defense	Heavy precipitation
Topic 4	Jiaozuo City	Jiyuan City	flood control	scenic spot	geologic hazard
Topic 5	Trapped	rescue	urgent	fireman	Guard peace
Topic 6	village	Danger	Man	Mountain torrent	search for and rescue
Topic 7	rescue sb.	Warm	Disaster situation	Emergency response	Man in White
Topic 8	be on duty	traffic police	rainstorm	Xinxiang City	Waterlogging

**Table 2 behavsci-13-00282-t002:** Text topic extraction results in the explosion period.

Explosion Period					
Topic 1	Flood	Save-help	Food	danger avoidance	Infectious Diseases
Topic 2	mutual aid	rescue	urgent need	material	contact number
Topic 3	urgent	retention	Shutdown	Rescue	Heavy rainfall
Topic 4	Sandbag	forward	Xinxiang City	volunteer	Rescue team
Topic 5	Help-seeking	Document	material	spread	Hotel
Topic 6	Donation	Charity	come on.	Stars	Hero
Topic 7	rescue	Ponding	Anhui Province	Jiangsu Province	fireman
Topic 8	typhoon	extreme	metro	rare	Heavy precipitation

**Table 3 behavsci-13-00282-t003:** Text topic extraction results in the recession period.

Recession Period					
Topic 1	missing	die in an accident	Disaster-affected	Agriculture	Infectious Diseases
Topic 2	fire control	reinforce	disinfection and sterilization	Shandong Province	Waterlogging drainage
Topic 3	urgent need	Waterlogging	Rescue team	Assault boat	Rainstorm mutual aid
Topic 4	Shutdown	rush to repair	Metro	Free	victim
Topic 5	typhoon	flood	be careful	early warning	Infectious Diseases
Topic 6	Policeman	traffic police	rescue	Hero	Village cadres
Topic 7	material	donation	rescue	urgent	“ERKE”
Topic 8	Rescue	material	public welfare	Fans	Rescue team

**Table 4 behavsci-13-00282-t004:** Text topic extraction results in the stabilization period.

Stabilization Period					
Topic 1	investigation	Acceptance	rainstorm	Zhengzhou City	Infectious Diseases
Topic 2	to guard against	Hedging	prediction	Strong convection	Emergency response
Topic 3	Mountain torrent	early warning	transfer	rescue	geologic hazard
Topic 4	Local areas	north	Continue publishing	Chinese Central Meteorological Station	Rainstorm blue warning
Topic 5	flood prevention	Shutdown	combat a flood	Sandbag	A new round
Topic 6	7.20	Stop full	extreme	one month	save against a rainy day
Topic 7	Road	citizen	vehicle	be prepared to meet the challenge	Rainstorm red warning
Topic 8	rainstorm	Damaged	reconstruction	Donation	“ERKE”

## Data Availability

The data supporting the findings of this manuscript are available from the corresponding authors upon reasonable request.

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
