# Peer review of "Dynamic Characteristics and Evolution Analysis of Information Dissemination Theme of Social Networks under Emergencies"

_behavsci, 2023, doi:10.3390/bs13040282_

Round 1

Reviewer 1 Report

This paper combines LDA, TF-IDF and PMI for topic modelling of social media data and analyzes the topic dissemination during an emergency. While I believe the work is interesting and useful, I got the following suggestion for authors’ consideration.

1.   The sentence from Line 12-14 is kind of ambiguous and misleading. The current expression looks like the study specifically investigates the Rainstorm Event in Henan, while study actually uses the event as a case study to draw some generalizable conclusion. I thus suggest the authors paraphrasing the sentence, like “This paper uses … as a case study, to……”, to make the focus of your study clearer.

2.   The authors better explain why the Henan Rainstorm Event is selected as case study, to justify your selection.

3.   There are some terms given without explaining the meaning first, such as theme coding, life cycle. I suggest the authors explaining these terms first in the introduction section before using them.

4.   I think the explanation of LDA (section 3.3) and TF-IDF (section 3.5) should be included in methods section, rather than results section.

5.   This paper uses a single case study to draw a general conclusion. I think the authors better justify how the conclusion of this case can be transferable and generable. If yes, why. If not, then this should be admitted as the limitation.

6.   The current referenced works seems not enough, only 27. I suggest expanding the referencing part by adding more relevant works. Here are some works I recommend the authors citing: “Categorisation of cultural tourism attractions by tourist preference using location-based social network data: The case of Central, Hong Kong”, “Analysis of the performance and robustness of methods to detect base locations of individuals with geo-tagged social media data”.

Reviewer 2 Report

The study is well organized and addresses a relevant topic. It offers both theoretical and practical contributions, but the practical contributions are little explored by the authors. The theoretical contextualization of the topic has several gaps and should be deepened.

Improvement suggestions:

- The Introduction section should be improve to explore the role of social media in dealing with emergencies.

- This sentence is very relevant but it needs to be cited “In the literature, researchers divide the whole event into different communication cycles and summarize the public opinion themes in different periods” Which studies?

- This observation is relevant “Solomon et al. [5] note that it is difficult to stop a pandemic when it breaks out, and the best way to do so is to identify danger signs as soon as they appear” There are many studies that confirm this vision. Try to support this observation in multiple references considering also the experience of Covid-19.

- Authors note “Compared with traditional text semantic analysis, LDA has a better effect in analyzing large-scale unstructured data sets [13]” Structured data sets are also relevant fo your study? Please justify it.

- The amount of data used by the authors can be considered as big data. However, big data concept is not presented considering the characteristics of the data.

- Authors note “In addition, five topic words with high relevance under each topic were selected” Please clarify your concept of high relevance?

- Clarify in the Conclusions section the practical contributions of this study.

- Authors note “In future research, we will consider collecting data from various platforms..” Which kind of platforms? Which kind of data.

- The number of references is relatively low. I recommend improving the theoretical context of the study. Particularly the Introduction sections needs to be improved and include more relevant references.

Round 2

Reviewer 1 Report

The authors have addressed my concerns

Reviewer 2 Report

The paper can be accepted at this phase.